# The Requirement of Turkey Herpesvirus (HVT) Glycoprotein C During Natural Infection in Chickens and Turkeys

**DOI:** 10.3390/pathogens14060538

**Published:** 2025-05-28

**Authors:** Huai Xu, Widaliz Vega-Rodriguez, Kathrine Van Etten, Keith Jarosinski

**Affiliations:** Department of Pathobiology, College of Veterinary Medicine, University of Illinois at Urbana-Champaign, Urbana, IL 61802, USA; huaixu2@illinois.edu (H.X.); widaliz.v@gmail.com (W.V.-R.); vanettn2@illinois.edu (K.V.E.)

**Keywords:** herpesvirus, turkey, chicken, transmission, Marek’s disease (MD), vaccines

## Abstract

The glycoprotein C (gC) of gallid alphaherpesvirus 2—better known as Marek’s disease (MD) virus (MDV)—and gallid alphaherpesvirus 3 is required for horizontal transmission in chickens. Since gC is conserved within the *Alphaherpesvirinae* subfamily, we hypothesized that gC was also essential for the horizontal transmission of meleagrid alphaherpesvirus 1 (MeAHV1) or turkey herpesvirus (HVT). To test this hypothesis, we generated a fluorescent protein-tagged clone of recombinant (r)HVT (vHVT47G), removed the open reading frame of HVT gC from the genome (vHΔgC), and rescued the deletion by inserting an HA-epitope tagged HVT gC (vHΔgC-R) to test their ability to transmit in chickens and turkeys. We also tested whether MDV gC could compensate for HVT gC during transmission, where HVT gC was replaced with MDV gC (vH-MDVgC). Although all viruses replicated in chickens, none spread from chicken to chicken. However, when tested in turkeys, all viruses except vHΔgC transmitted from turkey to turkey. Importantly, the rescuent virus (vHΔgC-R) and HVT expressing MDV gC (vH-MDVgC) rescued transmission, showing that HVT gC is required and MDV gC can compensate for HVT gC for turkey-to-turkey transmission. These data confirm the host-specific transmission of HVT in turkeys and suggest that the essential function of alphaherpesvirus gC proteins is conserved. This information can be exploited while generating future vaccines against MD that will affect the poultry industry worldwide.

## 1. Introduction

Marek’s disease (MD) stands out as a significant challenge in the poultry industry due to the tremendous economic losses it causes [1]. This disease is caused by oncogenic and virulent gallid alphaherpesvirus 2 or Marek’s disease virus (MDV)—species *Mardivirus gallidalpha2*—which triggers malignant T-cell lymphomas, paralysis, and immunosuppression. Meleagrid alphaherpesvirus 1, more commonly called turkey herpesvirus (HVT)—species *Mardivirus meleagridalpha1*—has been widely utilized worldwide as a live vaccine, either alone or in combination with other serotypes, to protect against MD caused by the virulent MDV [2]. MD vaccines do well to reduce tumor formation and disease but do not stop the horizontal transmission of virulent MDV, increasing virulence over the decades [3]. Therefore, understanding the transmission dynamics of MDV and MD vaccines is vital for developing vaccines to stop MDV from spreading in poultry houses.

HVT was isolated from turkeys and is believed to be widespread among commercial turkey flocks. It spreads efficiently throughout turkey flocks [4]. HVT is avirulent in chickens [5], which can induce viremia associated with the induction of protective immunity against MD. Since the early 1970s, the HVT-based vaccine has been effectively used to confer immunity against MDV infection. Additionally, non-pathogenic gallid alphaherpesvirus 3 (GaAHV3) SB-1 strain and attenuated MDV CVI988 Rispens vaccines have been employed since the 1980s and 1990s.

*Alphaherpesvirinae* conserved glycoprotein C (gC) is not required for cell-to-cell spread but is important for the initial attachment of cell-free herpesviruses to heparin- and chondroitin sulfate proteoglycans on the surface of cells [6,7,8] and the final stages of virus egress from cultured cells [7,9]. It also has immune evasion functions, as herpes simplex virus (HSV) 1 (HSV-1) and 2 (HSV-2), suid alphaherpesvirus 1 (SuAHV1) or pseudorabies virus (PRV), equid alphaherpesvirus 1 (EqAHV-1), and bovine alphaherpesvirus 1 (BoAHV-1) gC proteins bind and inhibit the action of complement component C3 [10,11,12,13,14]. Notably, the interaction of gC proteins with C3 is species-specific [12]. For chicken herpesviruses, MDV, and GaAHV3 (strain 301B/1 strain), gC is not essential for in vitro propagation but is required for horizontal transmission or natural infection in chickens [15,16]. This is consistent with the importance of gC for HSV-1 and VZV replication in human skin cells [17]. In addition, MDV, GaAHV3, and HVT produce membrane-bound (MgC) and secreted gC proteins (SgC104 and SgC145) during replication in the skin [18], with the secreted forms being the predominant mRNA expressed in feathers for MDV [19]. Importantly, all three forms of gC (MgC, SgC104, and SgC145) are required for the efficient transmission of MDV in chickens [20]. Thus, the role of gC proteins during natural infection is likely a complex mechanism, where gC may play multiple roles.

Former studies examining HVT transmission in turkeys and chickens suggest HVT can transmit from turkey to turkey, turkey to chicken, and chicken to turkey [5]. In contrast, transmission from chicken to chicken was unsuccessful; however, transmission efficiency studies have not been performed previously. We have formerly shown that the gC of MDV [15,20,21] and GaAHV3 MD vaccine strain 301B/1 [16] is dispensable for in vitro and in vivo replication, but is required for horizontal transmission from chicken to chicken. Here, we hypothesize that the absolute requirement of gC for MDV and MD vaccine 301B/1 horizontal transmission in chickens is conserved among other avian herpesviruses and their hosts. To test this hypothesis, we used an infectious bacterial artificial chromosome (BAC) clone of the MD vaccine strain HVT FC126 in experimental and natural infections in chickens and turkeys to determine whether HVT gC is required for HVT transmission. Our results demonstrate that HVT does not readily transmit from chicken to chicken but does efficiently transmit from turkey to turkey. Importantly, HVT gC is required for horizontal transmission, and MDV gC could compensate for HVT gC in this process. These results extend our findings that the role of gC homologs in horizontal transmission is a conserved function across different herpesviruses and hosts and suggest the importance of investigating this glycoprotein during the horizontal transmission of other herpesviruses.

## 2. Materials and Methods

### 2.1. Cell Cultures

All cells were cultured at 38 °C in a humidified environment with 5% CO_2_. Chicken embryo cells (CECs) were derived from 10 to 11-day-old White Leghorn specific-pathogen-free (SPF) embryos sourced from the University of Illinois at Urbana-Champaign (UIUC) Poultry Farm (Urbana, IL, USA) using established protocols [22]. Initially, primary CEC cultures were established in growth medium comprising Medium 199 (Cellgro, Corning, NY, USA) supplemented with 10% tryptose-phosphate broth (TPB), 0.63% NaHCO_3_ solution, antibiotics (100 U/mL penicillin and 100 μg/mL streptomycin), and 4% fetal bovine serum (FBS). Upon reaching confluence, CEC cultures were transitioned to Medium 199 supplemented with 7.5% TPB, 0.63% NaHCO_3_, 0.2% FBS, and antibiotics for maintenance.

### 2.2. Viruses

A BAC clone containing the genome of the HVT strain FC126 was obtained from Elanco Animal Health (Greenfield, IN, USA), and recombinant (r) HVT rHVT47G, rHΔgC, rHΔgC-R, and rH-MDVgC were generated previously [18]. Briefly, rHVT47G expresses UL47eGFP, which is used to track replication in feathers [23]. The UL44 (gC) open reading frame was deleted to generate rHΔgC; then, vHΔgC was used to create rHΔgC-R and rH-MDVgC, which express HVT gC with an N-terminal hemagglutinin (HA) epitope or MDV gC. Each rHVT was reconstituted by transfecting primary CEC cultures with purified BAC DNA plus Lipofectamine 2000 (Invitrogen, Waltham, MA, USA) using the manufacturer’s instructions, then further propagated in CEC cultures until virus stocks could be stored. All viruses were used at ≤5 passages for in vitro and in vivo studies.

### 2.3. Measurement of Plaque Areas

Plaque areas were assessed in CEC cultures following the method described previously [24]. Briefly, CEC cultures were seeded in six-well tissue culture plates and inoculated with 100 plaque-forming units. After five days, cells were fixed and anti-HVT chicken sera and goat anti-chicken IgY-Alexa Fluor^®^ 488 secondary antibody (Molecular Probes, Eugene, OR, USA) were used for immunofluorescence assays (IFAs). Digital images of 50 individual plaques were captured using an EVOS FL Cell Imaging System (Thermo Fisher Scientific, Hanover Park, IL, USA) and processed using Adobe^®^ Photoshop^®^ version 21.0.1. Subsequently, plaque areas were quantified using ImageJ [25] version 1.53d software. Box and Whisker plots were constructed, and Student’s *t* tests and a one-way analysis of variance (ANOVA) were included as the fixed effect to determine significant differences.

### 2.4. Viral Replication Kinetics in Cell Culture

Multistep growth curves were used to assess viral replication in the cell culture. CEC cultures were seeded in 6-well tissue culture plates and infected with 100 PFU/well of each respective virus the following day. To quantify viral genomic copies, total DNA was extracted from the inoculum (0 h) and infected cells at 24, 48, 72, and 96 h post-infection (pi) using the DNA STAT60 from Tel-Test, Inc. (Friendship, TX, USA). To measure the relative replication of rHVTs, the comparative CT method (2^−ΔΔCt^ method) was used in qPCR assays. HVT genomic copies were measured using primers targeting HVT ICP4 (Forward; 5′TGGCGGAGAACGATACGATG3′, Reverse 5′AAACGCGCATTGTCTGGAAC3′) and normalized to chicken/turkey vimentin (Forward; 5′GGAACAATGATGCCCTGC3′, Reverse; 5′GCAAAATTCTCCTCCATTTCAC3′). Each primer set was designed using the NCBI/Primer-BLAST program [26]. with settings of 100 to 250 nt for product length, 60 ± 3 °C for optimal Tm, and *Meleagris* for the organism. Primer sets were tested for specificity using qPCR assay parameters, and PCR products were gel-purified and sequenced to confirm specificity. The 2× Power SYBR^®^ Green PCR Master Mix (Thermo Fisher Scientific) was used with an Applied Biosystems QuantStudio 3 Real-Time PCR System (Thermo Fisher Scientific). The data were analyzed using the QuantStudioTM Design & Analysis Software v1.4.2 provided by the manufacturer. The fold increase over the inoculum was determined in triplicate for each virus at each time point to measure the relative viral replication.

### 2.5. Ethics Statement

All animal work was conducted according to national regulations and ARRIVE guidelines. The animal care facilities and programs of UIUC meet the requirements of the law (89–544, 91–579, 94–276) and NIH regulations on laboratory animals and are compliant with the Animal Welfare Act, PL 279. The Association for Assessment and Accreditation of Laboratory Animal Care accredits UIUC and the College of Veterinary Medicine at UIUC. All experimental procedures complied with the approval of UIUC’s Institutional Animal Care and Use Committee.

### 2.6. Animal Experiments

Pure Columbian SPF chicks were purchased from the University of Illinois Poultry Farm (Urbana, IL, USA), and standard SPF turkey poults were purchased from The Ohio State University. To test the replication of rHVTs in chickens and turkeys, 4-day-old chicks or poults were experimentally infected by an intra-abdominal inoculation of 10,000 PFU for vHVT47G, vHΔgC, vHΔgC-R, and vH-MDVgC and housed with uninfected naïve contact birds to measure horizontal transmission. Water and food were provided ad libitum for all animal experiments. Due to limited chambers, up to three rHVT were tested simultaneously. In chicken trials 1 and 2, and turkey trial 2, 8–10 chicks or poults were experimentally infected and housed with an equal number of uninfected naïve contact birds. In turkey trial 1, 5–7 turkey poults were used for experimental infection and naïve contact turkeys. Chicken trials 1 and 2 and turkey trial 2 were terminated at 8 weeks, while turkey trial 1 was terminated at 10 weeks.

### 2.7. Viral Replication Kinetics in Turkeys

A total of 40 microliters of whole blood were collected in 20 µL 0.1 M EDTA by wing-vein puncture from experimentally infected turkeys at 3, 7, 14, 21, and 28 days pi (n = 8–10/group) and then frozen at −80 °C until all samples were collected. DNA was extracted from 10 µL blood-EDTA using the E.Z. 96 blood DNA kit from Omega Bio-tek, Inc. (Norcross, GA, USA) according to the manufacturer’s instructions.

To measure the relative level of viral replication of HVT in turkeys, TaqMan qPCR was used with previously described primers and probe against HVT *sorf1* [27] and chicken *iNOS* [24,28] genes purchased from Integrate DNA Technologies, Inc. (Coralville, IA, USA). There is only a single nt difference between chicken and turkey *iNOS* gene primers; therefore, the chicken *iNOS* primers were used to normalize for turkey DNA. To generate standard curves in qPCR assays, the BAC DNA of rHVT47G and a plasmid containing the *iNOS* gene were used [28] in serial 10-fold dilutions, starting with approximately 500 pg DNA. The coefficient of regression for each qPCR standard curve was always >0.98. For qPCR, the reaction mixture contained Universal TaqMan Master Mix (Applied Biosystems, Foster City, CA, USA) with ~0.5 µg DNA, 25 pmol of each gene-specific primer, and 10 pmol of the gene-specific probe in 25 µL volumes. qPCR assays for HVT *sorf1* were performed in an ABI Prism 7500 Sequence Detection System (Applied Biosystems, Inc.), and the results were analyzed using Sequence Detection System v.1.9.1 software. iNOS qPCR assays were performed using the QuantStudio 3 or 7 Pro Real-Time PCR System (Thermo Fisher Scientific), and the results were analyzed using the QuantStudioTM Design & Analysis Software v1.4.2 supplied by the manufacturer. The average viral genomic copies per cell was determined by normalizing HVT *sorf1* copies to *iNOS* copies.

### 2.8. Direct Fluorescence and Immunofluorescence Assays (IFA) of Feather Follicles (FFs)

To monitor the timing of when each vHVT47G or its derivatives reached the feathers where HVT is shed, two flight feathers were plucked from both the right and left wings (totaling four feathers) of experimentally infected or contact-exposed chickens or turkeys every week.

To detect HaHVTgC and MDV gC in feather follicles, whole feathers were plucked from chickens infected with different vHVTs and fixed using PFA buffer (2% paraformaldehyde, 0.1% Triton X-100) for 15 min, washed twice with PBS, and then blocked in 10% neonatal calf serum (Sigma-Aldrich, St. Louis, MO, USA). Fixed FFs were stained with primary rabbit anti-HA (Cell Signaling Technology, Inc., Danvers, MA, USA) or mouse anti-gC A6 [29] antibodies with anti-rabbit or anti-mouse Ig Alexa Fluor 568 (Molecular Probes, Eugene, OR, USA) used as the secondary antibody. The Leica M205 FCA fluorescent stereomicroscope with a Leica DFC7000T digital color microscope camera (Leica Microsystems, Inc., Buffalo Grove, IL, USA) was used to analyze stained FFs. All images were compiled using Adobe^®^ Photoshop^®^ version 21.0.1.

## 3. Results

### 3.1. Replication of vHVTs in Cell Culture

Following the reconstitution of rHVT clones with UL44 removed (vHΔgC) and replaced with HA-tagged HVT gC (vHΔgC-R) or MDV gC (vH-MDVgC), we tested replication in CEC cultures using plaque size assays (Figure 1a) and multistep growth curves (Figure 1b). Plaque size assays showed no significant differences between all four viruses using Student’s *t*-tests (Figure 1a). However, virus growth kinetics showed differences at 4 days pi, where vH-MDVgC replicated significantly better than all three other viruses and vHΔgC replicated better than vHΔgC-R and vHVT47G. These results show that adding the HA epitope to the N-terminus of HVT gC did not affect HVT replication in the cell culture, and HVT gC was not essential for HVT replication in the cell culture.

### 3.2. Replication and Transmission of vHVTs in Chickens

To test our hypothesis that, like MDV gC and 301B/1, HVT gC would be required for the horizontal transmission of HVT, we first tested vHVT47G and vHΔgC in chickens. To follow replication, the expression of UL47eGFP was monitored in feathers over 8 weeks. Both viruses replicated in chickens following experimental infection, as evidenced by UL47eGFP expression in feathers, but both vHVT47G and vHΔgC did not transmit from chicken to chicken (Figure 2a).

Since vHVT47G did not transmit from chicken to chicken, we hypothesized that HVT expressing MDV gC (vH-MDVgC) could transmit from chicken to chicken. Therefore, we tested the ability of vH-MDVgC to replicate and transmit in chickens. Similar to vHVT47G and vHΔgC in trial 1, vHVT47G, vHΔgC, and vH-MDVgC did not transmit from chicken to chicken (Figure 2b). Direct fluorescence from UL47eGFP expression from infected chickens confirmed that all viruses replicated in the FFs.

For both trials, whole blood was collected from all contact chickens, serum was tested for anti-HVT antibodies using IFA, and blood was used to measure HVT DNA using qPCR assays. It was confirmed that all chickens negative for UL47eGFP in feathers were also negative for anti-HVT antibodies in their serum and viral DNA in their blood.

### 3.3. Replication and Transmission of vHVTs in Turkeys

Herpes viruses are typically host-specific. Therefore, we next tested the ability of HVT to transmit in turkeys. Not surprisingly, in our first trial, vHVT47G, vHΔgC, and the vHΔgC rescuent (vHΔgC-R), in which HVT gC with an HA tag was added back to vHΔgC, showed that both vHVT47G and vHΔgC-R were transmitted from turkey to turkey (Figure 3a). In contrast, vHΔgC could not naturally infect turkeys, suggesting that gC is also required for HVT transmission in turkeys. The IFA of FFs from infected turkeys confirmed the expression of HaHVTgC in the vHΔgC-R group.

Next, we tested the ability of MDV gC to compensate for HVT gC in HVT transmission in chickens in a second trial. In this experiment, vHΔgC could not transmit to turkeys again, while vHΔgC-R and vH-MDVgC could infect contact chickens; however, the transmission of vH-MDV was delayed compared to vHΔgC-R (Figure 3b). FFs stained with the anti-MDV gC antibody were negative for vHΔgC-R-infected FFs and positive for vH-MDV gC-infected FFs, showing that vH-MDV gC expresses MDV gC.

Whole blood in EDTA was collected from all remaining turkeys at termination, DNA was extracted, and qPCR was used to measure HVT DNA, where we confirmed that all chickens negative for UL47eGFP in feathers were also negative for viral DNA in their blood.

Since vH-MDVgC had delayed transmission, we measured viral genome copies in experimentally infected chickens in trial 2. There was no significant difference between all three viruses at 3, 7, 14, 21, and 28 days pi, except vHΔgC-R had significantly higher viral genome loads at 14 days pi compared to vHΔgC and vH-MDVgC (Figure 3c).

These data combined show that HVT efficiently transmits from turkey to turkey, HVT gC is required for turkey transmission, and MDV gC can compensate for this transmission.

## 4. Discussion

This report used the rHVT vector based on an FC126 vaccine BAC clone to study HVT transmission in chickens and turkeys. In addition, we tested the importance of gC for transmitting HVT in both birds. From our results, we could confidently conclude that (1) HVT does not efficiently transmit from chicken to chicken but does spread efficiently from turkey to turkey in our experimental/shedder model, (2) HVT gC is required for the horizontal transmission of HVT from turkey to turkey, and (3) MDV gC can compensate for HVT transmission in turkeys.

### 4.1. HVT Can Efficiently Transmit in Turkeys, but Not in Chickens

Studies in the 1970s suggested limited or no transmission of HVT from chicken to chicken [5,30,31,32]. However, controlled experimental studies were limited. Our experimental transmission model showed that although HVT replicated and reached the feathers of experimentally infected chickens with high efficiency, contact chickens remained uninfected (Figure 2). One limitation of our chicken trials was that they were only performed for eight weeks due to limited space to hold chickens as they grew. Thus, it is possible that transmission from chicken to chicken could occur over extended periods. Additionally, chickens were inoculated at 3 days of age, which is similar to studies by Cho et al. [32], which found no transmission from chicken to chicken when day-old to four-day-old chicks were inoculated with HVT and transmission was evaluated from 6 to 8 weeks. However, when eight-week-old chickens were inoculated, they found that HVT spread from chicken to chicken over 8 weeks. Thus, our studies are consistent with former studies examining the inoculation of young birds and evaluating transmission under 8 weeks.

### 4.2. HVT gC Is Required for HVT Transmission in Turkeys

Our studies primarily aimed to test the importance of the alphaherpesvirus conserved gC protein for the horizontal transmission of the MD vaccine strain HVT. In chickens, we could not conclude whether gC was required for transmission, as neither the wild-type HVT (vHVT47G) nor gC-null (vHΔgC) was transmitted (Figure 2). However, in turkeys, gC-null HVT (vHΔgC) could not transmit from turkey to turkey in two trials (Figure 3). In contrast, both wild-type (vHVT47G) and vHΔgC rescuent (vHΔgC-R), in which HVT gC was added back to vHΔgC to include an N-terminal HA tag, were able to transmit to contact turkeys. These results confirmed the essential role of HVT gC for the horizontal transmission of HVT in turkeys, consistent with other members of the *Mardivirus* genus in their host species [15,16,21].

### 4.3. MDV gC Can Compensate for HVT gC in Transmission in Turkeys

Former studies showed that HVT gC can compensate for MDV gC in facilitating the transmission of MDV in chickens [33]. Similarly, studies swapping GaAHV3 gC for MDV gC showed that MDV gC can compensate for GaAHV3 gC in the transmission of GaAHV3 in chickens [16]. In contrast, the infectious laryngotracheitis virus or GaAHV1, which is a member of the *Iltovirus* genus, could not compensate for MDV gC in transmission, showing that the function of gC in transmission is conserved among the *Mardiviruses*. Our data here further support these former findings. However, after numerous experiments in chickens, and now turkeys, although it appears that the gC proteins of MDV, GaAHV3, and HVT can be swapped and facilitate the transmission of the respective parental virus, typically, the transmission is less efficient or delayed compared to the parental gC protein, as can be seen in this report (Figure 3b). The reason for this is unclear. However, MDV, GaAHV3, and HVT all express full-length and alternatively spliced mRNA variants to produce secreted gC proteins [18]. We hypothesize that *Mardivirus* gC plays two significant roles during transmission, one in attachment to cells and the second in binding immune molecules during natural infection. This is based on our studies showing that MgC and SgCs can facilitate MDV transmission individually, but this is significantly reduced compared to viruses expressing both MgC and SgCs [20]. Thus, as more data on gC’s role in transmission for avian herpesviruses are generated, the overall theme suggests that gC is involved in multiple processes during natural infection. That is, although *Mardivirus* gC proteins can compensate for transmission, that suggests one function is conserved between the different gC proteins; other functions may be compensated for, resulting in a less efficient or delayed transmission. Further studies are warranted to identify these divergent functions, and the avian herpesvirus natural infection models can be used to address this question.

Combining the HVT vaccine with the CVI988 Rispens strain or the SB-1 strain has proven to be an effective bivalent vaccine strategy against more virulent MDV strains [34]. Other benefits of HVT vaccines include the capacity to genetically alter the virus to express protective heterologous antigens and the possibility of in ovo vaccination. Through targeted gene expression, rHVT vaccines are considered a versatile viral vector against several poultry disease-associated viruses due to some attractive features: (1) for chickens, it is a non-pathogenic, persistent infectious agent that causes an ongoing immune response that maintains elevated levels of protective antibody titers, (2) the HVT vaccine’s lyophilized form allows it to withstand long-term storage and transportation [35], and (3) the HVT genome is large enough to insert several foreign genes. HVT vectors expressing genes from the Newcastle disease virus, infectious bursal disease virus, and infectious bronchitis virus provide protective systemic immune responses against these pathogens [36]. Our findings that HVT expressing MDV gC (vH-MDV gC) does not affect and may enhance its replication (Figure 1b) in cell culture, replicates efficiently in chickens (Figure 2), but cannot facilitate chicken transmission, suggest that swapping HVT gC for MDV gC may enhance HVT protection against MD without affecting replication.

## 5. Conclusions

This report extends our understanding of the conserved role gC plays in host-to-host transmission to HVT in chickens and turkeys. Here, we concluded that HVT does not efficiently transmit from chicken to chicken but does spread efficiently from turkey to turkey; HVT gC is required for the horizontal transmission of HVT from turkey to turkey; and MDV gC can compensate for HVT gC in transmission in turkeys. Our results support our hypothesis that the absolute requirement of gC during horizontal transmission is conserved among the *Mardiviruses*. In addition, gC proteins can be exchanged between members of the *Mardiviruses*. One question to be addressed is whether HVT expressing MDV gC enhances protection against MD. This also begs the question of whether other HVT genes can be exchanged for MDV genes that could enhance HVT protection indices. Our studies form a strong baseline for addressing the challenges of generating the next generation of MD vaccines.

## Figures and Tables

**Figure 1 pathogens-14-00538-f001:**
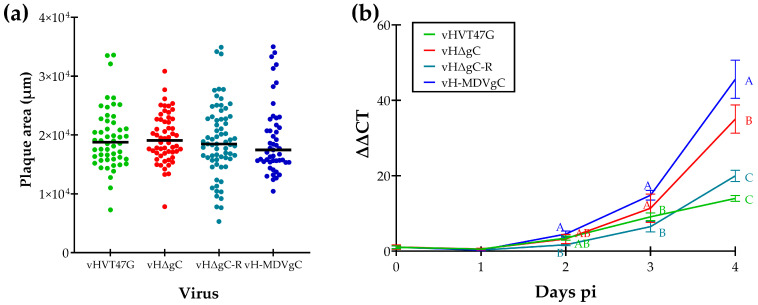
Replication of rHVT in cell culture. (**a**) Plaque areas for viruses reconstituted from vHVT47G, vHΔgC, vHΔgC-R, and vH-MDVgC were measured from 40 to 50 plaques per virus at 4 days pi, and the individual and mean areas (-) are shown. No significant differences were determined using the Student’s *t*-test. (**b**) Multistep growth kinetics were used to measure virus replication in CEC cultures. Viral DNA was extracted daily from infected cells in triplicate and subjected to a qPCR using gene-specific primers for the HVT *ICP4* and chicken vimentin genes as the internal control. The relative expression levels of the viral DNA were calculated using the comparative CT method (ΔΔCT method). The mean fold-change in viral DNA copies over the inoculum (day 0) is shown for each virus and time point. Averages with different letters (A–C) are significantly different using one-way ANOVA (*p* ≤ 0.05).

**Figure 2 pathogens-14-00538-f002:**
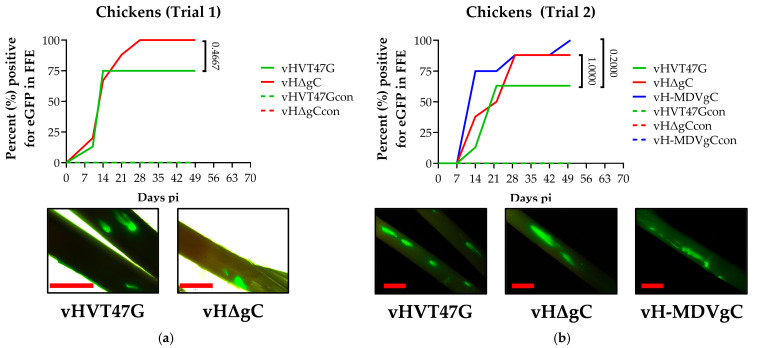
Replication of rHVT in chickens. Chickens were experimentally infected with vHVT47G, vHΔgC, and vH-MDVgC as described in the Materials and Methods, in two trials (**a**,**b**). Feathers were plucked from infected and contact birds over eight weeks, and the percentage of infected birds was enumerated over time. Fisher’s exact tests were used to determine significance, and *p* values are shown. Representative feathers for each group are shown using direct fluorescent microscopy for UL47eGFP at 28 days post-infection. The red line represents 2 mm.

**Figure 3 pathogens-14-00538-f003:**
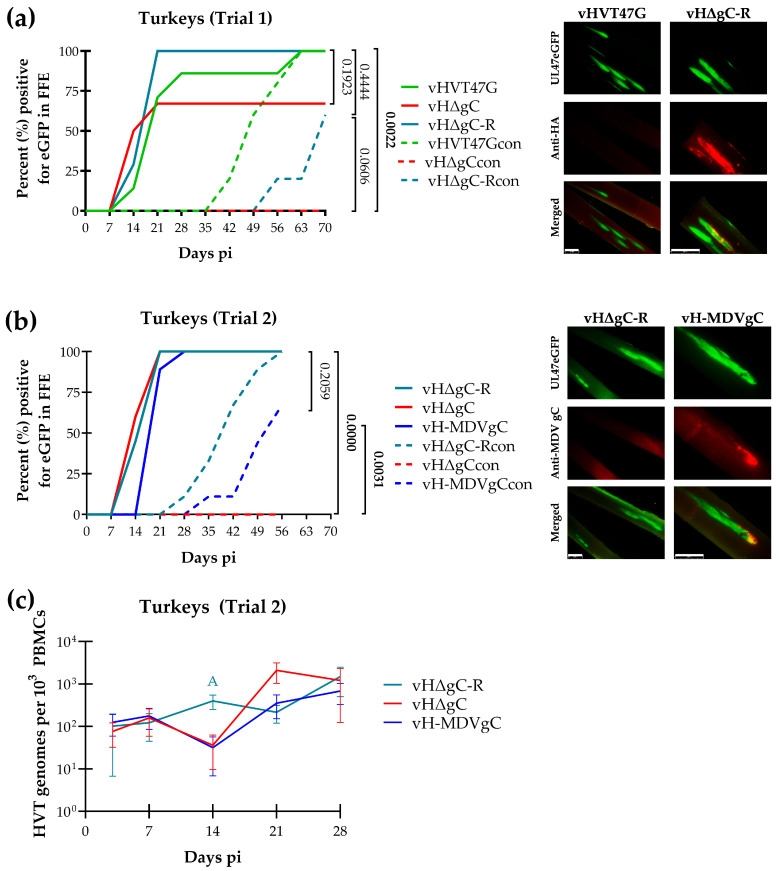
Replication and horizontal transmission of rHVT in turkeys. Turkeys were experimentally infected with vHVT47G, vHΔgC, vHΔgC-R, and vH-MDVgC, as described in the Materials and Methods, in two trials (**a**,**b**). Feathers were plucked from infected and contact (con) birds over ten weeks, and the percentage of infected birds was enumerated over time. Fisher’s exact tests were used to determine significance, and *p* values are shown. Significant differences are bold. Feathers were plucked from infected turkeys at 28 days pi, fixed, and stained using anti-HA (**a**) or anti-MDV gC (**b**) antibodies. Expression of UL47eGFP is shown by direct fluorescence, and HaHVTgC or MDV gC is shown using indirect immunofluorescence assays. The white bar in the merged image represents 2 mm. (**c**) Replication was monitored in experimentally infected turkeys by quantification of HVT genomes in PBMCs over the first four weeks of infection. The mean HVT genomic copies per 10^3^ blood cells ± standard error of means is shown. Two-way ANOVA was performed to check the significant effect of virus and time (pi) and the associated interaction effect (virus × time). Only vHΔgC-R has significantly higher viral genomes at 14 days than vHΔgC and vH-MDVgC, and is noted with an “A”.

## Data Availability

All data of this article are found within the article.

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
