# Peer review of "The Requirement of Turkey Herpesvirus (HVT) Glycoprotein C During Natural Infection in Chickens and Turkeys"

_pathogens, 2025, doi:10.3390/pathogens14060538_

Round 1

Reviewer 1 Report

Comments and Suggestions for Authors

The manuscript "The requirement of herpesvirus of turkeys (HVT) glycoprotein C during natural infection in chickens and turkeys" shows interesting results contributing to the knowledge of the HVT dynamics. It is generally well-written, however, as there are several experiment and conditions, I suggest to divide some sections and show the results in a clearer way by including tables.

1.I suggest authors to briefly describe the characteristics of each clone employed in this manuscript.
2.The dilutions used for the plaque assay are missing, please include them.
3.If the primers used for the HVT ICP4 PCR were designed for this study, more methodological information describing it should be included.
4.The 2.6 section in Materials and Methods can be confusing, I suggest to divide the animal experiments into subsections and include a Table in the results summarizing the main findings of these different conditions.
5.How were the chickens and turkeys euthanized?
6.Similarly, I suggest to include a Table for results of the viral replication kinetics experiment, for better clarity.
7.Authors should explain why the feather follicles were selected to monitor the infection and why other organs were not included in the study as well.
8.In the plaque assay, it is not clear if the differences were observed just in size or also in the number of plaques.
9.What does A, B and C mean in the Figure 1b and A in Figure 3c? please mention it in the figure caption.

Author Response

1.I suggest authors to briefly describe the characteristics of each clone employed in this manuscript.

Response: Thank you for this suggestion. We have included the following in section 2.2 (Lines 98-102): “Briefly, rHVT47G expresses UL47eGFP, which is used to track replication in the feathers [23]. The UL44 (gC) open reading frame was deleted to generate rHΔgC, then vHΔgC was used to create rHΔgC-R and rH-MDVgC that express HVT gC with an N-terminal hemagglutinin (HA) epitope or MDV gC. .”

2.The dilutions used for the plaque assay are missing, please include them.

Response: The information for performing plaque area assays is referenced. However, for simplicity, we included the following statement on section 2.3 (Lines 108-11): “Briefly, CEC cultures were seeded in six-well tissue culture plates and inoculated with 100 plaque-forming units. After five days, cells were fixed and anti-HVT chicken sera and goat anti-chicken IgY-Alexa Fluor® 488 secondary antibody (Molecular Probes, Eugene, OR) were used for immunofluorescence assays (IFAs).”

3.If the primers used for the HVT ICP4 PCR were designed for this study, more methodological information describing it should be included.

Response: Thank you for noticing this. We have included more details in section 2.4 (Lines 127-133); “Each primer set was designed using the NCBI/Primer-BLAST program [26] with settings of 100 to 250 nt for product length, 60 ± 3 °C for optimal Tm, and Meleagris for the organism. Primer sets were tested for specificity using qPCR assay parameters, and PCR products were gel-purified and sequenced to confirm specificity (data not shown).”

4.The 2.6 section in Materials and Methods can be confusing, I suggest to divide the animal experiments into subsections and include a Table in the results summarizing the main findings of these different conditions.

Response: The Materials and Methods section provides the general experimental plan, while the results provide more specific details.  We do not feel that giving a table summarizing the results would enhance the manuscript, as the Results and Conclusion sections summarize the main findings.

5.How were the chickens and turkeys euthanized?

Response: Chickens were euthanized using IACUC-approved asphyxiation using CO2.

6.Similarly, I suggest to include a Table for results of the viral replication kinetics experiment, for better clarity.

Response. We do not feel that showing viral kinetics in table form would help clarify things. The figures are intended to show the replication kinetics over time. No changes have been made to the manuscript.

7.Authors should explain why the feather follicles were selected to monitor the infection and why other organs were not included in the study as well.

Response: Thank you for this suggestion. Since we are studying transmission, we wanted to monitor the site where HVT is shed, the feather follicle epithelial cells. We have included the following on line 183: “…, where HVT is shed,…”

8.In the plaque assay, it is not clear if the differences were observed just in size or also in the number of plaques.

Response: Plaque areas only measure the area of individual plaques. As suggested above, we included more specifics in the assay to provide more details. Essentially, we infect cells with an equal number of plaque-forming units and then measure the area of plaques at 5 days. This is a standard method to measure replication of cell-associated viruses.

9.What does A, B and C mean in the Figure 1b and A in Figure 3c? please mention it in the figure caption.

Response: Thank you for noticing this. We neglected to include this in the legend.  We have added “(A-C)” in the legend on line 215 and included “Only vHΔgC-R has significantly higher viral genomes at 14 days than vHΔgC and vH-MDVgC, noted with an “A.” on lines 262-263.

Reviewer 2 Report

Comments and Suggestions for Authors

The manuscript discusses the virulence of herpesvirus for chickens and turkeys

I have detected some issues and these are outlines below.

Introduction. Please explain in detail the precise gaps in the literature that would be filled through this work.

Some passages from the Introduction can be moved to discussion to better support the ideas of the authors, without the text losing any important context.

M&M. There is a serious omission about the lack of description for controls. Please add a new section, to describe in detail all controls (strains, animals, consumables, procedures) used in this study.

Please add a timeline of the study with all the procedures carried out.

The viral replications kinetics must be explained in greater detail to help the readers understand the procedures with no reference to outside textbooks. This part of the manuscript must be greatly extended, I expect it to be at least one full page.

The lack of any statistical analysis is very concerning.

Results. The figures are very good, but should be increased in size.

Please add tables to summarise the findings and to make reading of the results easier.

Discussion. The Discussion is short and shallow. Please extend by adding further ideas in order to explain correctly the findings of the study.

As it is the Discussion now, it can lead to immediate rejection of the manuscript.

This lack of depth is reflected in the small number of references. I expect at least 80-90 references in the revised manuscript.

The Discussion must be divided into two sub-sections to allow easier flow of reading.

Conclusions. Please extend this section and please bring fully in line with the actual findings of the study.

Overall. The manuscript needs a significant revision and improvement. After resubmission, it should go again for peer review.

Recommendation. Major revision.

Author Response

 Introduction. Please explain in detail the precise gaps in the literature that would be filled through this work.

Response:  This is a hypothesis-driven report. Therefore, we state our hypotheses tested in this report (lines 72-74): “Here, we hypothesized that the absolute requirement of gC for MDV and MD vaccine 301B/1 horizontal transmission in chickens is conserved among other avian herpesviruses and their hosts.”

Some passages from the Introduction can be moved to discussion to better support the ideas of the authors, without the text losing any important context.

Response: In our view, the introduction is relatively short and do not feel a significant portion can be moved to the discussion without losing context. We focus on three overall subjects, Marek’s disease as the problem we are studying, HVT being used as a vaccine for Marek’s disease, glycoprotein C is the gene we are focusing on, and a paragraph about HVT transmission in chickens and turkeys that was studied almost 50 years ago for context.

M&M. There is a serious omission about the lack of description for controls. Please add a new section, to describe in detail all controls (strains, animals, consumables, procedures) used in this study.

Response: We are unsure of what the reviewer requests, as most of the information is provided in the context of each section. We neglected to include the chicken and turkey lines; therefore, we added the genetic strains of chickens and turkeys in the revised manuscript (Lines 87 and 146-147).

Please add a timeline of the study with all the procedures carried out.

Response:  We are unsure what the reviewer requests regarding a “timeline.”  The experimental “timeline” is found within Figures 2 and 3 as samples are collected. The X-axis is in days post-infection.

The viral replications kinetics must be explained in greater detail to help the readers understand the procedures with no reference to outside textbooks. This part of the manuscript must be greatly extended, I expect it to be at least one full page.

Response: Viral replication kinetics are a common technique used to measure virus replication in vivo. We have published our methods extensively and are referenced in this paper.  We have included some parts to make the overall kinetics more straightforward, including: “ To quantify viral genomes,…” (line 120) and “The average viral genomic copies per cell, as determined by normalizing HVT sorf1 copies to iNOS copies.” (lines 179-180).

The lack of any statistical analysis is very concerning.

Response: We are confused by this statement. We carefully describe statistical analysis in the Figure Legends. They can be found as follows:

Figure 1a (lines 209-210): “No significant differences were determined using the Student’s t-test.”

Figure 1b (lines 215-216): “Averages with different letters (A-C) are significantly different using one-way ANOVA (P ≤ 0.05).”

Figure 2 (lines 238-239): “Fisher's exact tests were used to determine significance, and P values are shown.”

Figure 3a,b (lines 253-254): “Fisher's exact tests were used to determine significance, and P values are shown.”

Figure 3c (lines 260-262): “Two-way ANOVA was performed to check the significant effect of virus and time (p.i.) and the associated interaction effect (Virus×Time).”

Results. The figures are very good, but should be increased in size.

Response: Thank you for this suggestion. We have increased the figures as requested.

Please add tables to summarise the findings and to make reading of the results easier.

Response: We believe this is a matter of preference. We do not feel that tables would fully summarize the transmission data, as the time to transmission is an important aspect to consider in transmission studies. We believe the XY plots of the data better summarize the transmission and replication results.

Discussion. The Discussion is short and shallow. Please extend by adding further ideas in order to explain correctly the findings of the study.

As it is the Discussion now, it can lead to immediate rejection of the manuscript.

This lack of depth is reflected in the small number of references. I expect at least 80-90 references in the revised manuscript.

The Discussion must be divided into two sub-sections to allow easier flow of reading.

Response: Thank you for your suggestions.  We have separated the discussion into three sections, focusing on our three major findings.

However, the number of references expected is unrealistic given the limited work on gC in transmission. Little work has been done on Marek’s disease and HVT transmission since the 1970s, and we reference these papers. There are many papers on human herpesvirus gC in cell culture, and we reference a few in our work. However, this manuscript is for the Special Issue on “Current Challenges in Veterinary Virology.” Therefore, we do not believe diving deep into human herpesvirus literature is warranted, apart from what we think is relevant to this work.

Conclusions. Please extend this section and please bring fully in line with the actual findings of the study.

Response: We have extended the Conclusion section to summarize the report better.

Round 2

Reviewer 1 Report

Comments and Suggestions for Authors

The comments were addressed by the authors

Reviewer 2 Report

Comments and Suggestions for Authors

All issues were resolved.